# A Census of Federally Regulated Big Cat Populations within the United States as of December 2020

## Rachel Garner

Garner Research & Consulting, Seattle, WA 98107, USA; contact@garnerconsulting.net

**Abstract:** No comprehensive assessment of the populations of big cats in federally regulated facilities in the United States is currently available. Concerns about big cat use are increasingly of strong public interest and the lack of data about their number and distribution has ramifications impacting zoological industry function, conservation programs, rescue work, and legislation. In this work a dataset has been compiled using publicly available USDA (United States Department of Agriculture) records and direct information requests. The resulting census was derived from the animal inventories listed on inspection records for all 2272 facilities with animal exhibition licenses. The total number of big cats in federally regulated facilities is on the order of 4100 animals and appears to be declining.

**Keywords:** big cat; Felidae; population biology

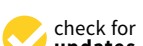



## 1. Introduction

Very little is known about the cumulative population of captive big cats in the United States. Although the public's desire for an understanding of commercial big cat use has recently skyrocketed due to major lobbying campaigns and media exposés such as the 2020 Tiger King documentary series [1], the need for quantification of populations of different big cat species within the country is not new. Understanding captive big cat populations in the United States can have more impact than just responding to public interest in these animals as the systemic and ongoing lack of data on captive cats impacts industry function, conservation programs, and legislative efforts.

As concerns about public safety and animal welfare have increased in the past decade, estimates of the population of tigers, as well as the numbers of all big cats held by private owners, have become common [2–6]. However, such often-cited estimates lack supporting data, and no reproducible attempts to verify those numbers have been published. Although some accrediting groups track the number of animals of a given species held by their members [7,8], these groups have not attempted to quantify the animals held by exhibitors outside of their membership bodies. Thus, no known comprehensive and reproducible attempts have been made to assess the documented captive big cat populations within the United States.

The American zoological industry is highly fragmented, resulting in a lack of knowledge among industry leaders about the overall population of big cat species within the US, as well as the carrying capacity of the zoological industry for those species as a whole. As facilities accredited by the Association of Zoos and Aquariums (AZA) face an increasing lack of space within their network to house animals as part of big cat conservation breeding programs [9], an understanding of what space and resources other sectors of the industry could contribute may assist with determining the future of those programs (as discussed in Powell [10]).

Many other entities, included but not limited to non-AZA-accredited zoos and big cat sanctuaries, have a vested interest in understanding the scope of the captive populations and how it impacts their ability to acquire animals. Smaller zoos accredited by the Zoological Association of America (ZAA), as well as non-accredited zoos, must often source large felids for display from populations external to the AZA Species Survival Plan (SSP).

With an estimated 500–700 zoological facilities in the United States [11,12], any facility interested in adding large felids to their collection or breeding big cats themselves needs to be aware of the size and availability of the current captive populations. Sanctuaries that house big cats should also have an interest in understanding not just the scope of private ownership, but of the commercial sector as well, as their inventories and therefore budgets are contingent on the number of animals in need of rescue or relocation. Accordingly, trends in any other sector of the exotic animal industry have a direct impact on needed future capacity and funding levels for rescue facilities.

Similarly, animal advocacy organizations working on ending the exploitation of exotic felids in the pet trade and commercial situations could use data about the movement of big cats into regulated sanctuaries to assess the impact of their campaigns. In addition, proposed legislation, rules, and ordinances regulating the ownership or management of big cat species frequently include exemptions that are specific to facilities maintaining a federal license and/or specific accreditations [13–15]. Thus, it would be highly beneficial for legislators to be informed about the number of constituents whose businesses would be impacted by their vote on these issues.

With increasing public scrutiny of the commercial use of big cats, it is crucial for all stakeholders to work from verifiable, reproducible data regarding the actual scope and distribution of the populations of those species. This paper aims to begin building a body of knowledge about the scope of big cat populations in the United States as a whole. Although it is difficult to collect data on privately owned animals, information on big cats held by federally regulated exhibitors is accessible through United States Department of Agriculture (USDA) records. In this paper, all big cat species inventories from available Animal and Plant Health Inspection Service (APHIS) inspection records are aggregated.

## 2. Materials and Methods

For the purposes of this study, the term "big cat" was defined in accordance with the prohibited wildlife species covered by the Captive Wildlife Safety Act [16] and the proposed Big Cat Public Safety Act [17], rather than phylogenetically. Big cats in this study therefore are considered to be tigers (*Panthera tigris*, all subspecies), lions (*Panthera leo*, all subspecies), cougars (*Puma concolor*), jaguars (*Panthera onca*), leopards (*Panthera pardus*, all subspecies), snow leopards (*Panthera uncia*), cheetahs (*Acinonyx jubatus*), clouded leopards (*Neofelis nebulosa*), and any hybrid of those species such as a liger or tigon.

All businesses in the United States that hold mammals while conducting commercial operations are subject to federal regulation. Nongovernmental entities that exhibit animals to the public in the course of routine business are required to maintain a Class C USDA license (colloquially, an exhibitors' license) through the Department of Agriculture [18]. The most recent routine inspection record for every currently active USDA Class C license [19] was accessed using the USDA-APHIS Animal Care Search Tool.

For this analysis, when multiple facilities were held under one USDA license, each was considered to be an individual entity, as they are inspected individually and each facility has a distinct business purpose requiring licensure. The animal inventories from each inspection were then searched for the presence of big cat species, and the location, species, and number of individuals were recorded. Some exhibitors had inspections for animals both in a home facility inspection and in travel status within the same year. As USDA records do not identify individual animals, it was not possible to determine if animals in travel status had already been counted through a home facility inspection earlier in the year. If a facility had both a home inspection and travel inspection with inventories that contained big cats from within the same inspection period, the home inventory was used and the travel inventory ignored. If an exhibitor only had travel-status inspections on the USDA-APHIS database, the most recent travel inspection report was used.

Some federally regulated entities known to be holding big cats did not have inventories available. This occurred due to procedural issues, record-keeping or database issues, or because a facility was government-run and therefore is not required to maintain a

USDA license. Inspection records undergoing appeal with the USDA are not online until resolved and exhibitors whose licenses have been suspended are not contained in the active exhibitors list, so facilities in either situation during early January 2021 would not have had their most recent inspection record available online. In addition, some facilities known to have big cat collections did not have any inspection records on the Animal Care database. Some exhibitors had routine inspections where the inventories did not include all species in their collection or had not had an inspection since USDA-APHIS began including animal inventories on inspection records. Where possible, data for exhibitors whose inventories were incomplete, missing, or unavailable were acquired by requesting a current inventory of the target species directly from the facility.

During data analysis, the decision to exclude one Washington, DC entity from the census was made due to the high probability that it was a duplicate; it shared the same inspection date, inspector, and animal inventory as the most recent inspection record for a business owned by the same individual in Texas.

The data were sorted by state, business type, accreditation, species, and number of animals held. The accreditation status and business purpose of each entity holding big cats was determined by using publicly available information. As USDA does not sub-divide licensees by business type, and industry usage of the relevant terminology varies broadly, definitions for business categories had to be determined to sort the data. Seventeen categories were identified (using definitions from the author's previous work, with a few modifications [20]) and each exhibitor was then assigned to one or identified as an unknown business type (as listed in Table 1). The two federally run facilities in the dataset were categorized by their business purpose despite the absence of USDA licensure.

**Table 1.** USDA Class C Licensed Exhibitor Business Categories.

| Category | Definition |
|---|---|
| Alternate Mission Nonprofit | A nonprofit organization for which animal-related causes are not the primary mission but for which animal exhibition is a component |
| Amusement Park | An amusement park that includes an animal exhibition component |
| Camp/Resort | A vacation- or camp-style facility whose guest activities require them to maintain an exhibition license |
| Conservation Breeder | A business that breeds animals specifically for conservation programs that maintains an exhibition license for commercial support of the program |
| Educational Institution | A university or other place of learning with an animal collection for the purposes of student teaching and/or research that also maintains an exhibition license to make the collection available to the public |
| Entertainment/Outreach (animal-reliant) | A business that facilitates animal-based entertainment, education, or outreach programs held away from where the animals reside (e.g., birthday parties with animal presentations) |
| Entertainment/Outreach (not animal-reliant) | A business that does not exist for the purposes of animal exhibition that maintains an exhibition permit to include animals in their programming (e.g., magicians) |
| Mascot | A college or university that maintains an exhibition license for the purpose of having a live animal mascot |
| Museum | A permanent education facility that includes an on-site animal collection for exhibition and education programs |

**Table 1.** *Cont.*

| Category | Definition |
|---|---|
| Nature Center | A facility specifically dedicated to educating the public about local ecosystems and wildlife that includes an on-site animal collection |
| Rescue/Rehabilitation | A wildlife rescue and rehabilitation facility that offers tours and on-site education programs with unreleasable animals |
| Production Work | A company that uses their animal collection in the pursuit of artistic projects (e.g., movies, theater, photography) |
| Park | A non-business area set aside for public recreation with an on-site animal component |
| Petting Zoo | A facility that allows the members of the public to interact directly with domestic, farm, and small exotic animals on-site |
| Retail Services | A business where animal exhibition is a supporting factor of primary business objectives (e.g., steak house, pet supply store, bed-and-breakfast) |
| Sanctuary | A business that provides a stationary collection of rescue animals with a permanent home and maintains an exhibition license to allow the public to visit or for commercial support of the facility |
| Zoological Facility | A business that maintains a stationary collection of exotic animals for the primary purpose of public exhibition |
| Unknown | Not enough information about this business is available for classification |

## 3. Results

### 3.1. Data Range

The date range for the most recent listed inspections of all active Class C licenses (hereto referred to as "federally regulated facilities" or "exhibitors") was between 29 March 2014 and 30 December 2020. The date range for the most recent listed inspections of all active exhibitors holding big cats was between 28 June 2017 and 17 December 2020. Most of these inspections were conducted during or since 2018. Inspection records for six facilities known to house big cats were either incomplete or unavailable; inventories provided by those respective facilities were accurate to the dates 16 February 2021 (two facilities), 17 February 2021, 22 February 2021, and 30 July 2021 (two facilities).

### 3.2. Federally Regulated Businesses Holding Big Cats

Out of 2226 active licensed exhibitors and two federally run entities, this census identified 448 facilities holding at least one big cat (approximately 20%). Zoological facilities accounted for the majority (62%, 279) of the exhibitors, as shown in Figure 1, followed by sanctuaries (17%, 76). The rest of the business types categorized had far fewer facilities in each category; the total for all 16 other business types and the unknown exhibitors comprised 21% of the total number of regulated businesses (93).

Among the federally regulated facilities holding big cats, approximately 40% (176) were accredited by one of the two main zoological accrediting bodies in the United States (AZA and ZAA) that cover terrestrial mammals. Figure 2 breaks the data down further by accreditation: 135 facilities were accredited by only the Association of Zoos and Aquariums (AZA) (30% of all exhibitors), and 33 were accredited by only the Zoological Association of America (ZAA) (7%). Another 8 facilities were accredited by both AZA and ZAA (2%) (It should be noted that these numbers include 13 non-zoo facilities accredited by either or both the AZA and ZAA.) The remaining 116 zoological facilities were unaccredited by either organization (26%). The majority of the 76 sanctuaries holding big cats were also unaccredited. The rest of the business types categorized had far fewer facilities

with 7 facilities accredited by the Global Federation of Animal Sanctuaries (GFAS) (2%), 7 facilities accredited by the American Sanctuary Association (ASA) (2%), and 6 facilities accredited by both (1%).

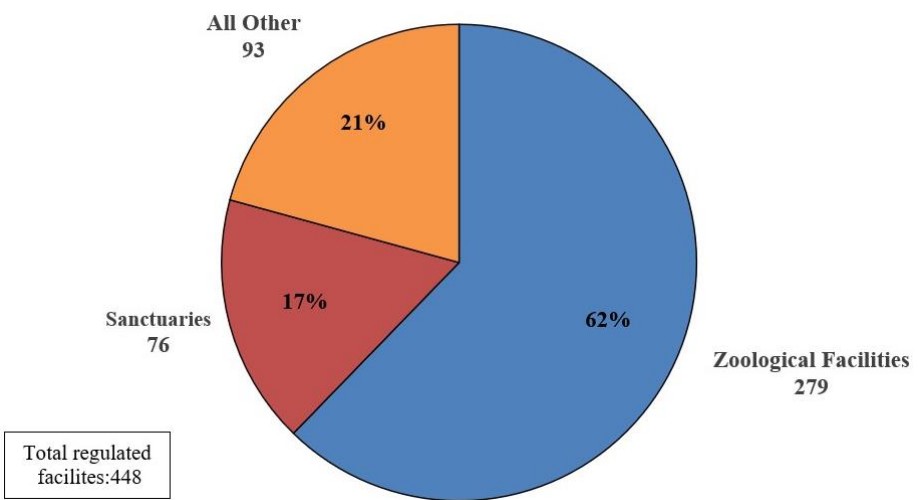

**Figure 1.** Federally regulated facilities holding big cat species, sorted by business type.

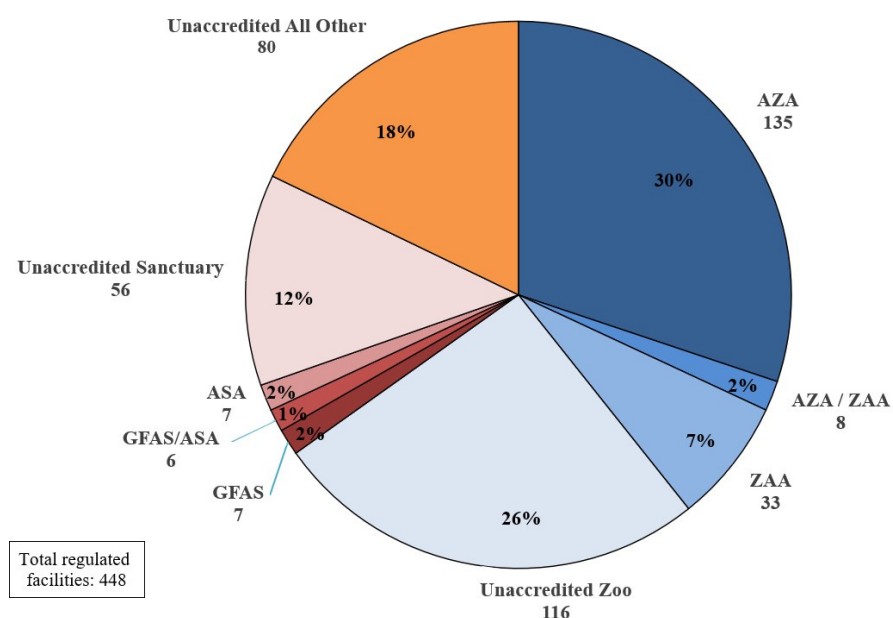

**Figure 2.** Federally regulated facilities holding big cat species sorted by accreditation. *Note: Both the AZA and the ZAA accredit a small number of non-zoo facilities.*

*3.3. Big Cat Population Distributions*

4103 big cats were identified across all federally regulated business types. As shown in Figure 3, the most common species was tigers, with a total of 1538 individuals across all business types. The next most populous was lions, with 862, followed by 487 cougars and 425 cheetahs.

As shown in Figure 4, the slight majority (45%) of big cats are held by unaccredited facilities (including unaccredited zoos, sanctuaries, and other identified business types). A total of 42% of big cats are held by zoological facilities that are either accredited by AZA or ZAA, and 13% of the total big cat population is held by sanctuaries accredited by GFAS or ASA.

The greatest number of big cats were found in Florida California, and Texas respectively (Figure 5). Wyoming, Delaware, and Vermont were recorded as having none.

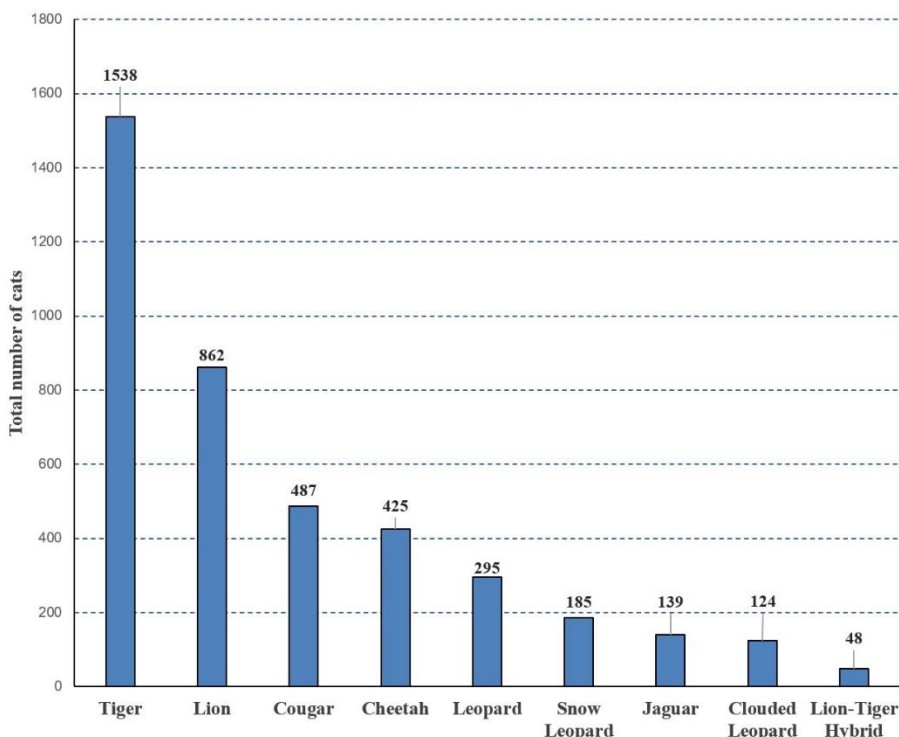

**Figure 3.** Total population of each big cat species within all federally regulated facilities.

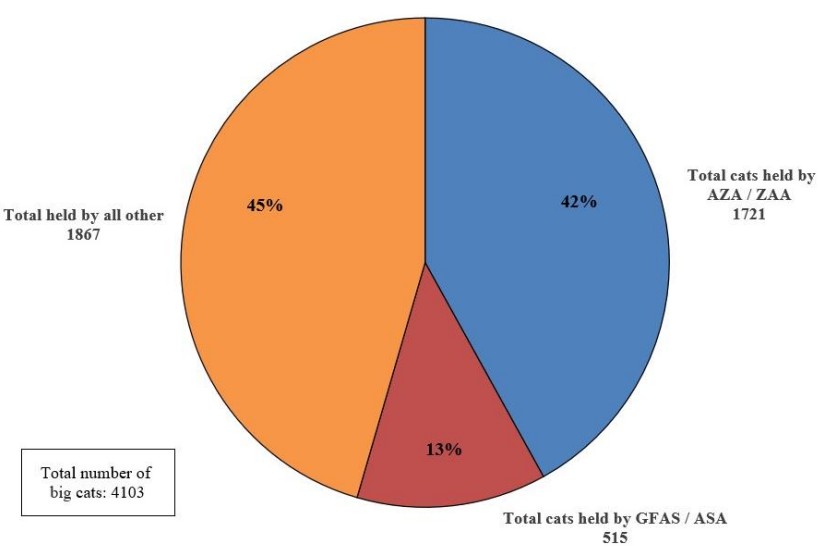

**Figure 4.** The total number of big cats in federally regulated facilities, sorted by facility accreditation.

The distribution of entities holding big cats was more uniform than the distribution of the big cats themselves, although the high and low ends of the range were similar (Figure 6): Florida had the most, followed by California and Texas, and Wyoming, Delaware, and Vermont had none.

### 3.4. Tiger King Animals

The timing of this study required that the animals involved in Tiger King be specifically addressed, as only the portion of Joseph Maldonado-Passage's tigers that moved to a sanctuary in 2017 [21] and those owned by Bhagavan Antle were included in this census.

Many of the people who were the subjects of the series had either a suspended or a forfeited USDA license during the time that the USDA records for this current census were collected [22,23], resulting in the exclusion of their inventories. The data collection period of this census occurred, in all but one case, after animals were seized from the Tiger King facilities but before the facilities that received them underwent another USDA inspection. (The last seizure occurred during after data collection was complete.) To ensure those big cats were counted, non-USDA sources such as press releases, media articles, and publications from receiving institutions were used to identify and track the locations of as many animals as possible. In total, the Tiger King animals comprise 111 big cats that are not represented in the census for either the state they originated in or the states they now reside in.

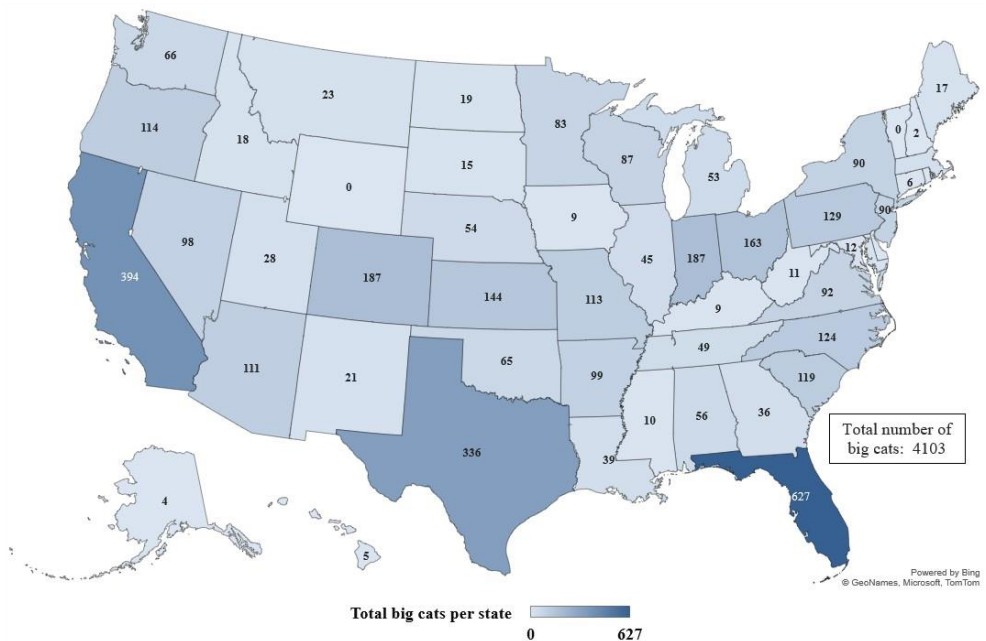

**Figure 5.** Total number of big cats in federally regulated facilities in each state.

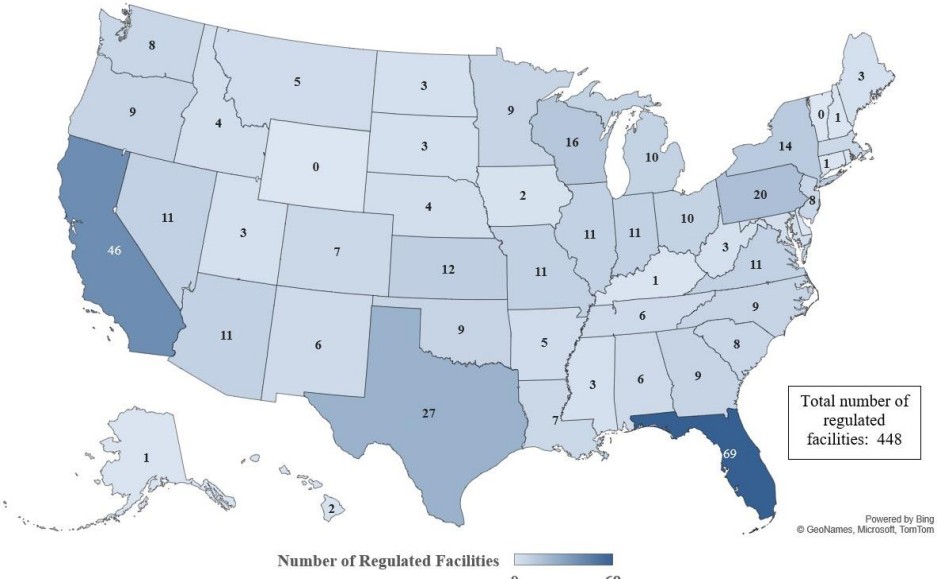

**Figure 6.** Total number of federally regulated facilities holding big cats in each state.

Twenty-two big cats were seized from Tim Stark/Wildlife in Need on 18 September 2020, as part of the requested relief for an Endangered Species Act citizen lawsuit brought

by the People for the Ethical Treatment of Animals (PETA) [24]. These animals would have been counted in the data for the state of Indiana in this census, but Stark had lost his USDA license in August of 2020 [23] and as such was not included on the list of active licensees used for data collection. Seventeen of those big cats (five tigers, eight lion–tiger hybrids and one lion, all housed at Wildlife in Need, as well as three lions owned by Stark but housed by Jeff Lowe) were transported to The Wild Animal Sanctuary in Colorado [24,25]. The other eight (four tigers and four lions) were transported to Turpentine Creek Wildlife Refuge in Arkansas [26]. Neither group of cats is represented in the census data, as the receiving sanctuaries' most recent inspections, as of the period data collection was completed, were in January and April of 2019, respectively.

Jeff and Lauren Lowe abandoned three tigers when vacating the Greater Wynnewood Exotic Animal Park property in Wynnewood, Oklahoma on 2 October 2020; the tigers were taken to The Wild Animal Sanctuary in Colorado [27]. Fourteen tigers were seized from the Lowes' "Tiger King Park" property in Thackerville, Oklahoma on 24 January 2021; these tigers were also taken to the Wild Animal Sanctuary in Colorado [28]. An additional 69 big cats were seized from the same property on or before 20 May 2021, in accordance with an order by the Department of Justice [22]. Of those 69 cats, there were 46 tigers, 7 lions, 1 jaguar, and 15 lion-tiger hybrids [29]. None of these cats are represented on the census for the state of Oklahoma since the Lowes' USDA license was suspended in February of 2020 and voluntarily terminated on 21 August 2020, and they never obtained a USDA license for the Thackerville location [22]. The Thackerville site was required to provide a complete inventory to the Department of Justice during the judicial case that resulted in the removal of their animals, but those documents are not available through the USDA search tool. What is known from media articles and social media posts by receiving facilities is that seven cats (6 tigers and 1 lion) were moved to the Keepers of the Wild sanctuary in Arizona [30]; 2 tigers were moved to Lions, Tigers, and Bears in California [31]; Turpentine Creek Wildlife Refuge in Arkansas took 13 cats (1 jaguar, 1 lion-tiger hybrid, 1 tiger; other species counts unknown) [32]; and The Wild Animal Sanctuary in Colorado took 14 tigers and 35 big cats (species counts unknown) [29]. It has not been publicly shared where the other 11 animals from the Thackerville location were moved.

## 4. Discussion

### 4.1. Data Accuracy and Specificity

Prior to this study, the only efforts at conducting a comprehensive census of regulated big cats were undertaken by Culver [33] and Chambers [34] through the Feline Conservation Foundation (previously the Feline Conservation Federation). Werner [35] addressed the total number of large felids in the United States with his 2005 census, although he only specified numbers by species and location for tigers. The methodologies published for these three studies were incomplete, and in some cases the records used were not made publicly available. Therefore, the accuracy of these previous censuses cannot be verified. Other attempts to track big cat populations have been undertaken [36,37], but these studies have either not been made public, or were published in such a way that the accuracy of the underlying datasets or methodologies cannot be verified.

Although this census improves on the methodology of those previous attempts, it is inherently limited by the scope of federal regulation and what records are publicly available. As such, even this census cannot be taken as an exact count of all big cats kept in regulated United States facilities. Rather, this study is most useful as a snapshot of the approximate population of each species and their distribution across business types and geography. However, without being able to request and efficiently receive an up-to-date inventory from every licensed exhibitor in the country, our methodology is the best (and potentially only) way to obtain a view of the breadth of these populations.

Importantly, the dataset collected for this study is reflective of what falls under USDA oversight and what inventories were recorded by inspectors, although some inspections have obvious issues with missing species or duplicate animals. When it was possible to

identify such issues, facilities were contacted for corrected information, and it is likely that similar errors were not eliminated when inconsistencies within inspection records could not be identified.

Since the USDA inspects facilities on a rolling and risk-based schedule, periods as long as three years may pass between inspections at lower-risk facilities, and others considered low-risk may only be inspected when a complaint about them is received [38]. There is no publicly available information regarding the inspection frequency for any specific exhibitor, aside from what can be observed from the dates on their previous inspection reports. It is likely, therefore, that some cats have been duplicated in this census due to the timing of animal transfers between facilities. Animals that were born or have died since a facility's most recent USDA inspection would also not have been included.

There are several known facilities with large numbers of big cats that are not USDA-licensed and therefore do not have USDA inventories available: sanctuaries that focus on large cat species and are never open to the public, such as Tiger Haven in Tennessee and Pride Rock in Texas, do not need USDA licenses and therefore have no publicly available inventories. Thus big cat populations in some states are known to be undercounted.

The data from this census were not validated against studbook data due to a high probability of error that could not be corrected for; studbooks and breeding and transfer plans for the AZA Species Survival Plans (SSPs) are updated regularly, and the data they contain will therefore only match the USDA records for a given facility if it were inspected close to the publication date of those documents. These documents are not updated in real time when breeding or transfer plans are carried out, so they do not reflect births, deaths, or transfers until the next program update. The ZAA tracks populations of only one big cat species, the cheetah, as part of voluntary participation in their Cheetah Master Plan Program (MPP) [7]. There does not appear to be any parallel effort to track the number of individual animals in big cat sanctuaries. Although some validating data might have been collected through Species 360 Zoological Information Management Software (ZIMS), not all facilities holding big cats use that information tracking software, so that database as well only reflects a portion of federally regulated big cats.

### 4.2. Data Limitations

The dataset addresses only the number of federally regulated big cats: it cannot be used to determine the number of privately owned big cats within the United States, nor the total number of all big cats in all situations within the country. Neither of the previous censuses published by the FCF could be used to derive such a number, as they were also narrower in scope than that work requires, although Chambers, 2016 can be misconstrued as addressing it. Werner did provide numbers claimed to represent the total of privately owned tigers and the total of privately owned big cats of all species in the United States, but he published no methodology for how those numbers were determined [35]. Many claims about the total populations in the United States cite tiger numbers as high as 5000+, and total big cat numbers of 10,000 to 20,000 for all species across all situations, but these estimates remain unverifiable [39]. The only actual census or published study that has been identified as a source for those claims was Werner's 2005 census [40], but Werner's published work did not include specific data sources or methodology.

### 4.3. Historical Context

The results of this census and of the previous censuses conducted by the FCF cannot be directly compared due to inconsistent or missing methodologies in the previous studies. Data categorized by species in the previous studies is likely to be more accurate than data categorized by business type due to the vagaries of earlier business classification. The reason for this is that Chamber's 2016 census split business types into zoos, sanctuaries, and "all other" (which appears to include non-USDA facilities as well as all other Class C licensees) [34,41], while Culver's 2011 census [33] looked only at USDA-licensed entities but subdivided the results into "AZA facilities only" and "all other USDA licensees." In

contrast, the species data assessed by both censuses is not subject to the same variability. Thus the population numbers produced by those censuses (Table 2), while known to not be wholly accurate, can still be used to suggest general trends in big cat populations and provide some context for the results of this study.

**Table 2.** Comparative species totals from this study and previous big cat censuses.

| Year | 2021 | 2016 [1]<br>(Chambers) | 2011 [2]<br>(Culver) | 2005<br>(Werner) |
|---|---|---|---|---|
| Number of facilities | 448 | 548 | N/A | N/A |
| Tiger | 1538 | 2330 | 2621 | 4692 |
| Lion | 862 | 1046 | 1024 | N/A |
| Cougar | 487 | 674 | 880 | N/A |
| Cheetah | 425 | 386 | 270 | N/A |
| Leopard | 295 | 367 | 562 | N/A |
| Snow Leopard | 185 | 164 | 170 | N/A |
| Jaguar | 139 | 140 | 141 | N/A |
| Clouded Leopard | 124 | N/A | 88 | N/A |
| Hybrids [3] | 48 | 37 | 32 | N/A |
| All | 4103 | 5144 | 5788 | 8977 |

[1] The FCF has indicated that the numbers in the 2016 census were inflated by Chambers, as he believed the species counts in his dataset were under-reported. [2] The original data from the 2011 census by Culver differ from the 2011 results that were reported in the 2016 census by Chambers. We have chosen to use the results from the original 2011 publication. [3] In both the 2016 and 2011 census data, the only hybrids explicitly included were ligers. It is unclear if non-liger hybrids were counted in these censuses.

*4.4. Population Observations*

The data shown in Table 2 allow general trends within the federally regulated big cat populations to be identified. Tiger populations within the US appear to have dropped drastically since 2011, by a total of perhaps about 1000 individuals. Lion and leopard populations have decreased in that timeframe as well, by perhaps approximately 200 individuals each. Captive cougars appear to have decreased in number by about 50%, while captive cheetah and snow leopard populations appear to have increased and jaguar numbers appear to have held steady. There are not enough comparative data available for clouded leopards from previous censuses to identify a trend, and the sample size for lion-tiger hybrids is so small that no real trend can be determined from the data that exist. One can speculate about the origins of many of these trends using historical context and industry data, although more research would be necessary to confirm them.

The apparent decrease in the tiger population may have been caused by a combination of factors: (1) the impact of rescue and the sanctuary sector on reducing the overall breeding population, (2) the cessation of major commercial breeding operations by USDA and/or the efforts of advocacy groups, (3) increasingly restrictive state laws regarding the private ownership of dangerous exotic species, and (4) changes to population management programs by the AZA. In 2009, the AZA Felid Taxon Advisory Group recommended that generic (subspecies hybrid or unknown lineage) tigers be managed to extinction to redirect resources towards purebred subspecies management programs; in the 12 years since, the generic tiger population within AZA institutions has declined from 114 to 30 animals [42].

It is unclear what caused the net 50% decrease in the cougar population in regulated facilities. However, unlike all other non-hybridized big cats addressed in this census, there is no formal collaborative breeding program for cougars within the country. The cougar population within facilities accredited by the AZA has been non-breeding since the late 1990s [43], so the only increase in population for this species within that portion of the zoological sector is through rescue of injured, abandoned, or nuisance wild cougars; according to the SSP data, an average of 8.5 animals enter their accredited institutions annually [43]. It is unknown at this point in time if data exist regarding how many cougars enter non-AZA facilities annually, or at what rate they reproduce in those settings.

Jaguar, snow leopard, and cheetah populations appear to have held steady or increased slightly over the last decade. It is possible that this is due to a larger fraction of their populations being part of SSP programs (53% of jaguars, 65% of snow leopards, and 65% of cheetahs are held by AZA-accredited institutions), compared to other species more commonly held in non-zoo facilities, where any increase in a captive conservation breeding population might not be noticeable due to attrition from other parts of the population.

Several other variables likely had some influence on the trends of all species covered by this census, but their impact cannot be determined without further research: (1) Some species of big cats are harder to breed in captive settings than others, and reproduction rates are also influenced by exhibit design and animal management practices. (2) Changes in the regulation and scrutiny of importation of endangered or CITES-listed species over the last decade are likely to have had an impact as well. (3) It is unclear the extent to which the known shortage of spaces for SSP animals within the AZA in general [10,44,45], and specifically for many of the large felid species covered by this census [8,9,46,47], may have also influenced population trends; while most of these programs need to increase their populations through breeding and importation of new genetics, the lack of available space to house offspring may discourage breeding. (4) The trend towards naturalistic exhibit design that houses a smaller number of animals per capita than historical designs with large amounts of holding may also be impacting big cat population trends as zoological facilities renovate their habitats.

### 4.5. Population Trend Validation through Rescue Records

Although censuses can suggest general trends in big cat populations over time through a series of snapshots, more comprehensive and regular tracking is required to enhance understanding and accuracy of these measures. Anecdotal evidence from the sanctuary sector implies, however, that changes to legislation and regulation in the past two decades have resulted in a significant decrease in the number of big cats found in non-zoo business types and non-accredited zoological institutions. Big Cat Rescue, a major big cat sanctuary that has coordinated the rescue and placement of big cats across the country for more than 25 years, noted in their 2020 annual report that requests for help have dropped drastically in the past 20 years [48]. In 2020 the facility received requests to place only four individuals of the species covered by the census, three of which were the tigers abandoned at the GW Zoo by the Lowes. The annual report did not include other Tiger King confiscations, as placement for those animals was determined by PETA and the United States Department of Justice respectively [23,49], but a review of the data provided by Big Cat Rescue indicates a shift from high double-digit numbers of big cats in need of rescue placement in the 2000s and 2010s to just a handful in each of the last few years [48]. Many high volume commercial big cat breeders have been shut down or have been forced to close in the last 20 years [50–52]. The four non-sanctuary entities involved in Tiger King (GW Exotic Animal Park/Joseph Maldonado-Passage; Tiger King Park/Jeff Lowe; Wildlife in Need/Tim Stark; Myrtle Beach Safari/Bhagavan Antle) were some of the last major commercial breeders in the country at the time the series aired; three of the four have since lost possession of all their big cats [22,25,49,53]. The results of this census reinforce how few USDA-licensed facilities with inventories capable of bulk-breeding exist today: it identified only ten non-accredited, non-sanctuary facilities with more than ten individuals of any single species of big cat. Sixteen facilities accredited by the AZA and/or the ZAA were found to have over ten individuals of at least one species; of these, only nine had over ten cheetahs.

### 4.6. Geographic Observations

The distribution of big cats across the United States identified from this census matches known historical, legal, and industry operation patterns. Almost all species have the highest density in California, Florida, and Texas. These three states have the highest concentration

of USDA exhibitors holding big cats, so it follows that they would also hold the highest number of most species.

Cougars are the species with the widest distribution across all states; as a native cat whose populations have been growing, and subsequently increasingly dispersing eastward [54,55], this is likely because facilities take in injured, abandoned, or nuisance animals that cannot be released to the wild.

Snow leopards are held almost exclusively by facilities in the northern two thirds of the United States. The data indicate that southern facilities, including those in Texas and Florida, generally do not house these animals. This may be due to the difficulty in providing them an appropriate climate and/or outdoor access during hot weather.

*4.7. Tiger King Animals*

Due to the unique timing of this census, 111 animals owned by individuals featured in Tiger King were not counted in the official census: 72 tigers, 15 lions, 1 jaguar, and 23 lion-tiger hybrids. Assuming all animals were moved to federally regulated facilities (the final locations of 11 of the big cats seized from the Lowes have not yet been made public), these animals will all be accounted for in the next USDA inspection of each facility.

## 5. Conclusions

Based on publicly available records, there are approximately 4103 big cats being held in approximately 448 federally regulated facilities within the United States. Most of the facilities holding big cats are zoological institutions, followed by sanctuaries, together accounting for 79% of all big cat populations. In addition, 111 big cats from the Tiger King facilities were identified that were excluded from the dataset because they were not listed as part of any licensed facility's inspection inventory during the data collection period.

Tigers are the most populous big cat species within the country, with 1538 individuals; lions and cougars are the next most common with 862 and 487 animals, respectively. There are less than 200 individual snow leopards, jaguars, clouded leopards, and lion-tiger hybrids.

An overall apparent decrease in regulated captive big cat populations was observed when the results of this census were compared with those of previous estimates. Tigers and cougars appear to have experienced the largest percentage decrease in populations; jaguar, snow leopard, and cheetah populations appear to have held steady or to be slightly increasing.

Some speculation regarding the origin of these trends was provided; however, further research is needed to study the factors influencing population trends in captive big cats in the United States. In the interim, this census provides a baseline which future studies can build upon to increase the general body of knowledge about how these species are held, bred, and transported.

**Funding:** This research was funded by the Feline Conservation Foundation. The analysis and conclusions of the study are solely those of the author.

**Institutional Review Board Statement:** Not applicable.

**Informed Consent Statement:** Not applicable.

**Data Availability Statement:** The dataset derived from this study is owned by the funding body. The Feline Conservation Foundation will consider reasonable requests to share the data.

**Acknowledgments:** The author would like to thank L. Tallman, J. Ramirez-Garofalo, N. Sakich, and E.C. Garner for their feedback and assistance in editing this paper.

**Conflicts of Interest:** The author declares no conflict of interest. Funding for this study was contingent upon submission of the results to an academic journal. The board chair of the funding organization facilitated data categorization of a few regulated businesses by requesting additional details about their reason for being USDA licensed. In addition, the same person attempted to facilitate the acquisition of data by requesting that several entities cooperate with the census effort.

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
