# Peer review of "A Census of Federally Regulated Big Cat Populations within the United States as of December 2020"

_2673-5636, doi:10.3390/jzbg2040037_

Round 1
Reviewer 1 Report
There are several concerns about the manuscript. The biggest concern in the discussion does not discuss the results. There are several new results introduced into the discussion. There are a few paragraphs in the discussion that would be better in the materials and methods subsequently strengthening the overall paper.
This paper probably has interest to readers of the journal as it attempts to quantify big cat numbers in the U.S.A. However, it need considerable revision to be of publishable quality. I have attached a copy with suggested revisions that will strengthen the paper.

Reviewer 2 Report
The article presents interesting data on the subject and is generally well written. I have written some remarks in the text, but I have some general remarks about the text :
- call each of the USDA-regulated exhibitors a population, since the numbers of individuals can be very different from one to the next. Probably also have a series of intervals of individuals or have a metric about it.
Materials and Methods
- There is some reason to consider this time interval between March 29, 2014 and December 30, 2020.
- Figures are not referenced in text, I don't know if it is part of the journal format, but usually the results are described and shown in the figure.
- The zoological facilities (line 133) do not mention in the text of the methods that this analysis is going to be performed and everything talks about the part of the license and the analysis by state, but never if they are accredited or not, and if they are accredited by any association. This can also be used as an issue in the discussion of who has to give the accreditation and if criteria should be homologated.
Discussion
- In the discussion part they should indicate who is to regulate all persons who have possession of an individual of any species of cat.
- Besides proposing to keep a clear register (studbook) of the cats that are in possession of each person. I understand that the regulation goes in the sense of having pure individuals, but not in regulating the possession of these organisms. In addition to all that implies the purchase and sale of these species.
- In the discussion there are several sections that are interesting but very descriptive, in some of these you will have to summarize more information and there are some things that seem to be repeated.

Round 2
Reviewer 1 Report
Greatly improved manuscript. It contains information that should be useful to readers.